# Acquisition and Exaptation of Endogenous Retroviruses in Mammalian Placenta

**DOI:** 10.3390/biom13101482

**Published:** 2023-10-04

**Authors:** Sayumi Shimode

**Affiliations:** 1Genome Editing Innovation Center, Hiroshima University, Higashi-Hiroshima, Hiroshima 739-0046, Japan; sshimode@hiroshima-u.ac.jp; 2Graduate School of Integrated Sciences for Life, Hiroshima University, Higashi-Hiroshima, Hiroshima 739-8526, Japan

**Keywords:** endogenous retrovirus, placenta, retrotransposon, Syncytin, Suppressyn

## Abstract

Endogenous retroviruses (ERVs) are retrovirus-like sequences that were previously integrated into the host genome. Although most ERVs are inactivated by mutations, deletions, or epigenetic regulation, some remain transcriptionally active and impact host physiology. Several ERV-encoded proteins, such as Syncytins and Suppressyn, contribute to placenta acquisition, a crucial adaptation in mammals that protects the fetus from external threats and other risks while enabling the maternal supply of oxygen, nutrients, and antibodies. In primates, Syncytin-1 and Syncytin-2 facilitate cell–cell fusion for placental formation. Suppressyn is the first ERV-derived protein that inhibits cell fusion by binding to ASCT2, the receptor for Syncytin-1. Furthermore, Syncytin-2 likely inserted into the genome of the common ancestor of Anthropoidea, whereas Syncytin-1 and Suppressyn likely inserted into the ancestor of catarrhines; however, they were inactivated in some lineages, suggesting that multiple exaptation events had occurred. This review discusses the role of ERV-encoded proteins, particularly Syncytins and Suppressyn, in placental development and function, focusing on the integration of ERVs into the host genome and their contribution to the genetic mechanisms underlying placentogenesis. This review provides valuable insights into the molecular and genetic aspects of placentation, potentially shedding light on broader evolutionary and physiological processes in mammals.

## 1. Introduction

Retroviruses possess RNA as their genome. During their life cycle, they convert genomic RNA to DNA using a reverse transcriptase and incorporate genomic DNA into the host genome using an integrase. When retroviruses are integrated into the germline genome DNA, they form an inherited proviral sequence called an endogenous retrovirus (ERV) [1,2,3]. Although most ERVs are inactivated by mutations, deletions, and epigenetic regulations (DNA methylation and histone modification), some retain transcriptional activity and play a role in host physiological functions, such as placentation, embryogenesis, and muscle cell differentiation [1,2,3], as well as in immune tolerance and resistance to infection by exogenous virus via a receptor interference mechanism [4,5,6,7,8].

The placenta, which comprises the endometrium from the mother and the trophoblast from the fetus, supports fetal life. After fertilization, the fertilized egg repeatedly divides, eventually becoming a blastocyst, which attaches itself to the uterine wall. The outer layer of the blastocyst called the trophectoderm ultimately forms the placenta. In humans, starting at embryonic day 6, mononuclear cytotrophoblasts fuse to form syncytiotrophoblasts [9]. The syncytiotrophoblasts interact and exchange substances with maternal blood, secrete placental lactogen and chorionic gonadotropin, and participate in immune tolerance. The morphology and structure of the placenta vary among animal species [10], and ERVs and retrotransposons are expected to be implicated in this diversity [10,11,12,13]. Syncytin proteins are encoded by modified *env* genes of ERVs. The Syncytin family, identified in various animals, was linked to placental dysfunction observed in preeclampsia, intrauterine growth retardation (IGR), and Down syndrome [14,15,16,17,18,19]. In this review, we aimed to consolidate the existing knowledge on the evolution of placental morphogenesis and physiology.

## 2. Evolutionary History of the Mammalian Placenta and Retrotransposons

### 2.1. Evolutionary History of the Mammalian Placenta

Mammals are divided into three major groups: monotremes, marsupials, and eutherians. Monotremes, which are the most primitive mammals, are ovoviviparous, although they still nourish their offspring with milk from the mother [20]. Most marsupials have a placenta that develops from the yolk sac; however, this placenta is relatively inefficient, resulting in a short gestation period (between 8 and 42 days) [21], leading to their offspring being born immature and raised in the marsupium [22,23]. Although not mammals, some fishes, reptiles, and amphibians have yolk sac placentas. Among them, hammerhead sharks have a long gestation period of approximately one year [24]. Eutherians additionally possess a chorioallantoic placenta [25]. In this review, we specifically discussed chorioallantoic placentas.

Approximately 250 million years ago (MYA), in the early Mesozoic era (Triassic), the oldest mammal, *Adelobasileus cromptoni*, diverged from reptiles [26]. Adelobasileus is believed to have been an oviparous species, living a slender existence amidst thriving dinosaurs [26]. Today, mammals inhabit diverse habitats worldwide [27]. Their survival and success can be attributed to the presence of placentas within the female body, which facilitates offspring development. Conversely, dinosaur and reptile mothers lay their eggs externally and provide minimal parental care, relying on laying numerous eggs for survival [28]. However, some mammals have evolved to possess placentas, allowing the mother to nourish her unborn child in her womb while continuing with her life activities. This adaptation offers protection from external factors, including weather threats and other exogenous risks, enabling the mother to invest more in a smaller number of offspring while ensuring their reliable growth [29].

In 2011, a fossil of a small animal resembling a shrew, measuring approximately 10 cm in length, was discovered in Liaoning, China. This animal lived 160 MYA (Jurassic period) and was named “Juramaia,” meaning Jurassic mother, owing to its possession of a placenta [30]. When the placenta is formed, part of the oxygen the mother takes in is provided to the child, with the placenta consuming 40% of the oxygen supplied to the fetoplacental unit [31]. During the Triassic era, the Earth was very dry, with sparse vegetation and an oxygen concentration of approximately 11% [32,33]. However, during the Jurassic period, the Earth warmed, vegetation flourished, and oxygen levels rose to 20% [32,33]. It was precisely during this period that mammals acquired placentas.

### 2.2. Placenta Acquisition and Retrotransposons

Retrotransposons, transposons, and repetitive sequences comprise approximately 40% of the eukaryotic genome [34]. Transposons relocate their genome using a cut-and-paste mechanism. In contrast, retrotransposons duplicate their genome: the retrotransposon RNA is reverse-transcribed, and the resulting DNA is inserted into another part of the genome. Moreover, retrotransposons contribute to the acquisition of the mammalian placenta. Monotremes diverged at 166 MYA [35]; then, paternally expressed gene 10/sushi-ichi retrotransposon homolog 1 (PEG10/SIRH1) was inserted into the common ancestor of marsupials and eutherians. Subsequently, marsupials diverged at 148 MYA, and PEG11/SIRH2 (retrotransposon-like 1, [Rtl1]) was inserted into the common ancestor of eutherians [36,37,38]. Both PEG10 and PEG11/RTL1 encode proteins that are highly homologous to group-specific antigen (GAG) and polymerase (POL) proteins of the sushi-ichi retrotransposon of the pufferfish genome; sushi-ichi retrotransposons belong to the gypsy retrotransposon family [39,40]. PEG10 knockout mice lack labyrinthine and trabecular chorionic layers and experience early embryonic lethality [41]. The Gag protein of PEG10 forms virus-like particles that bind to cellular RNAs, including Hbegf (Heparin-binding EGF-like growth factor), playing an important role in placental formation [42]. PEG10 might bind to and stabilize these cellular RNAs and was suggested to be involved in placentation [42]. Furthermore, mice deficient in the protease activity of PEG10 exhibit defects in placental fetal capillaries, resulting in perinatal lethality due to poor placental and fetal growth after mid-life [43]. The biochemical function of the Pol protein of PEG10 remains unknown; however, its protease domain is highly conserved in all Eutherians, suggesting that the protease activity plays an important role [43]. PEG11 is expressed in the endothelial cells of placental fetal capillaries, and PEG11 knockout mice experience embryonic lethality due to growth delay caused by disruption of the fetal capillary network [44,45]. Recently, Sirh7/Ldoc1 was also discovered as a retrotransposon-derived placenta-associated gene with a high homology to the GAG of the sushi-ichi retrotransposon. In Sirh7/Ldoc1 knockout mice, trophoblast differentiation and maturation are abnormal, and the expression and timing of hormones such as placental progesterone and placental lactogen from trophoblast giant cells are altered, resulting in abnormal pregnancy maintenance and delayed parturition [46]. Sirh7, as well as PEG11, was inserted into the common ancestor of eutherians [46]. These findings suggest that PEG10 is essential for primitive placenta formation and that PEG11 and Sirh7 promote the construction of eutherian-specific placental structures [36,41,46,47]. The biochemical functions of PEG11 and Sirh7 are also unknown and require future study.

## 3. Classification of the Mammalian Placenta

The placenta comprises trophoblastic cells from the fetus and uterine cells from the mother. Trophoblastic cells constitute the chorionic villi, which are divided into four types according to their distribution: diffuse, cotyledonary, zonary, and discoid placentas [10] (Figure 1). A diffuse placenta comprises chorionic villi that are scattered diffusely over almost the entire placental surface. A cotyledonary placenta is characterized by numerous ridges called caruncles in the endometrium; the chorionic villi invade to form circular patches called cotyledons. A zonary placenta takes the form of a belt around the fetus. Primates have discoid placentas, which appear as a round disk in a part of the uterus. Although primate placentas occupy a smaller area of the fetus than other placentas, the bond between the mother and fetus is typically the strongest.

Placenta classification is based on how deeply the trophoblast cells penetrate the endometrium of the uterus; a phenomenon termed “invasiveness”. Arranged from the most invasive to least invasive, placentas are hemochorial, endotheliochorial, syndesmochorial, and epitheliochorial [48] (Figure 1). Although exceptions exist, the diffuse, cotyledonary, zonary, and discoid placentas roughly correspond to the epitheliochorial, syndesmochorial, endotheliochorial, and hemochorial placentas, respectively [11]. Apes have a hemochorial placenta, in which cells on the placental side penetrate deeply into the endometrium, and the endometrium forms a decidua [11]. The placental villous epithelium is in direct contact with maternal blood, allowing the efficient exchange of maternal IgG antibodies, oxygen, and other substances [49]. Although the risk of miscarriage is relatively low with a hemochorial placenta, it takes longer for the placenta to detach during parturition, leading to more blood loss during this process [50]. Apes are social, and non-mothers in their community often participate in child-rearing, enabling them to assist each other during challenging births or when the afterbirth is damaging.

Physiological and morphological evolution, which are not consistent with the molecular phylogenetic tree (often referred to as convergent evolution), are also caused by the horizontal and inter-species/individual transmission of ERVs or retrotransposon copies via virion-producing copies from ERV families or other “carrier” viruses [51,52,53,54,55]. To date, multiple ERVs involved in placental diversification have been identified.

## 4. ERVs and Placenta

ERVs are retrovirus-like sequences that were previously integrated into the host germline genome. ERVs are passed on to offspring by vertical transmission via Mendelian inheritance [56] and spread between species and individuals by horizontal transmission [54]. ERVs were identified in all eukaryotes, constituting approximately 10% of the genome [34]. Most ERVs are inactivated by mutations, deletions, or epigenetic regulation; however, some retain their transcriptional activity and contribute to host physiology [3]. Several ERVs were identified to function in the placenta [11].

### 4.1. Syncytins

Cell–cell fusion is the fusion of cell membranes and the mixing of the cytoplasm to form multinucleated cells [57], which is essential for reproduction, development, and homeostasis in multicellular organisms [58,59,60,61,62]. The fusion of sperm and oocytes requires fertilization; during skeletal muscle development, myoblasts fuse and become multinucleated. In human and mouse placentas, mononuclear cytotrophoblast cells fuse continuously with overlying syncytiotrophoblast cells to maintain their function and exchange materials [63]. The ERV-derived Syncytin family is a cell–cell fusion factor with an important physiological role [60,64,65]. Moreover, retroviruses form syncytia [66,67,68]. Cell fusion is inhibited by antibodies against SU, a viral surface protein, suggesting that SU is essential for cell–cell fusion [69].

The fertilized egg undergoes repeated cell division and becomes a blastocyst, implanting itself into the uterine wall. The inner cell mass of the blastocyst forms the fetus, and the trophectoderm forms the placenta (Figure 2). The placenta, which comprises the endometrium derived from the mother and trophoblastic cells derived from the fetus, supports the life activities of the fetus. The placenta physically supports the fetus in the uterus and secretes hormones to maintain normal pregnancy. Syncytiotrophoblasts are located on the outermost side of the fetus and are formed by the fusion and multinucleation of cytotrophoblasts. Syncytiotrophoblasts interact and exchange substances with the maternal blood, secrete placental lactogen and chorionic gonadotropin, and are involved in immune tolerance [9,70].

Syncytins have functional roles in the placenta and are encoded by various ERV families. In humans, Syncytin-1 is encoded by the HERV-W *env* gene located on chromosome 7q21.2 [71,72]. Syncytin-1 mRNA is expressed in syncytiotrophoblasts, exhibiting stable expression levels throughout the gestation period [73]. Syncytin-2 is derived from the *env* gene of HERV-FRD and is located on chromosome 6p24.1 of the human genome [74]. Syncytin-2 is expressed in cytotrophoblasts, exhibiting decreased expression levels toward late gestation [73,75]. The Syncytin-1 receptor, alanine, serine, cysteine transporter 2 (ASCT2) is expressed in cytotrophoblasts [76,77,78,79], whereas the Syncytin-2 receptor, major facilitator superfamily domain containing 2 (MFSD2), is expressed in syncytiotrophoblasts [80]. Syncytin-1 and -2 comprise 538 amino acids, including surface (SU) and transmembrane (TM) subunits, and both possess cell–cell fusion ability [71,74]. Syncytin-2 has immunosuppressive activity, whereas Syncytin-1 does not [81]. During pregnancy, the mother harbors the fetus, "the stranger," which carries antigens of paternal origin. During egg laying, the mother is not exposed to antigens of paternal origin; however, placental acquisition necessitates the evolution of immunosuppressive mechanisms. The differences between Syncytin-1 and -2 regarding localization, expression level transition, and immunosuppressive activity suggest that they were acquired for different purposes. Syncytin-1 and -2 are found in primates, whereas Syncytin-A and B are found in mice and Syncytin-Rum1 in cattle [82,83,84]. Syncytin-A and -B were identified via an in silico search of the mouse genome [82]. The mouse placenta consists of three layers: trophoblast giant cells composed of cytoprophoblasts and syncytiotrophoblast layer-I and -II (ST-I and ST-II) composed of syncytiotrophoblasts. Syncytin-A and -B are expressed in ST-I and ST-II, respectively [85]. Syncytin-A KO mice are unable to form ST-I, and the fetuses die between embryonic days 11.5 and 13.5 [83]. Syncytin-B KO mice fail to form ST-II, but the fetuses survive, showing only developmental delay and a reduced number of offspring [86]. Syncytin-B fusion ability was confirmed in only canine MDCK cells. Syncytin-B has an immunosuppressive capacity and may have other functions than cell fusion [81]. Syncytin-Mab1 was also found in viviparous reptiles (Mabuya skinks) [87]. All syncytins have different origins, suggesting that retroviruses infected each animal lineage independently and acquired similar functions by chance. Furthermore, Syncytin was associated with cell–cell fusion during osteoclastogenesis [64] and cancer development [88,89,90].

### 4.2. Suppressyn

The two main routes of retrovirus entry into host cells are direct membrane fusion and endocytosis [91]. When the envelope protein of the retrovirus binds to a specific receptor on the cell surface, the envelope protein undergoes a conformational change, and direct membrane fusion occurs on the plasma membrane. In endocytosis, membrane fusion occurs in early endosomes after transport into the cytoplasm. EIAV, JSRV, and other retroviruses enter the host cells via endocytosis [92,93]. Receptor interference occurs when retroviruses using the same molecule as a receptor antagonize each other via competitive binding to the receptor. The endogenous avian leukemia virus (ALV) in chickens and the endogenous Jaagsiekte sheep retrovirus (JSRV) (enJSRV) in sheep function as antiviral factors against exogenous ALV and JSRV, respectively, via receptor interference [4,5,6]. The retroviral envelope protein comprises two subunits: SU, which contains a receptor-binding site, and TM. Many ERVs functioning as infection defense factors, such as enJSRV, inhibit infection via exogenous retroviruses by expressing the SU region rather than the full length of the *env* gene and by binding to the receptor [7]. Thus, ERV-derived antiviral factors were identified in various animal species, except in humans, for a considerable time. However, a human ERV-derived infection defense factor was recently discovered [8].

Suppressyn, the first ERV-derived protein shown to inhibit cell–cell fusion, is derived from the *env* gene of the HERV-H family of viruses (ERVH48-1) and is located on chromosome 21q22.3 of the human genome [94]. Northern blot analysis revealed that Suppressyn mRNA is expressed specifically in the placenta [75]. Single-cell RNA sequencing data from the placenta at multiple developmental stages showed that Suppressyn is expressed in cytotrophoblasts, extravillous trophoblasts, and syncytiotrophoblasts [8,12]. Immunostaining in the first trimester (at 7 weeks) showed that Suppressyn is expressed in cytotrophoblasts (especially in the progenitor and intermediate cytotrophoblasts) and extravillous trophoblasts but is limited in syncytiotrophoblasts [79]. Immunostaining in the second trimester (at 21 weeks) showed that Suppressyn is expressed in syncytiotrophoblasts and also in cytotrophoblasts [8]. Immunostaining in the third trimester showed that Suppressyn is expressed in syncytiotrohpblasts and extravillous trophoblasts [8,94]. Furthermore, immunostaining of blastocysts showed that Suppressyn is expressed in the trophectoderm and some Octamer-binding transcription factor 4-positive cells [8]. Suppressyn consists of SU sites with a stop codon preceding the SU-TM cleavage site. Suppressyn uses ASCT2 as a receptor to inhibit the Syncytin-1-mediated fusion of cytotrophoblasts by binding to ASCT2 [79,94]. Suppressyn is detected in cultured human trophoblast cells and placenta tissue samples in both intracellular and secreted forms, both of which bind directly to ASCT2 [94]. Both of them bind directly to ASCT2, suggesting that suppressyn might inhibit Syncytin-1-mediated cell–cell fusion via autocrine and paracrine mechanisms [94]. Intracellularly, Suppressyn alters the glycosylation of ASCT2 [79,94], resulting in changes in the ligand binding capacity [95] and protein stability [96] of ASCT2. However, the glycosylation changes in ASCT2 do not affect the cell fusion mediated by Syncytin-1 [94]. ASCT2 is an amino acid transporter that transports glutamine, alanine, serine, cysteine, and threonine in a Na+ ion-dependent manner [97] and is expressed in tissues throughout the body. Particularly, glutamine is essential in the placenta, and inhibiting its uptake leads to impaired embryonic development [98]. ASCT2 functions as a receptor for the feline ERV RD-114 virus, baboon ERVs, and simian retroviruses, among others. These viruses which use ASCT2 as a receptor are referred to as the RD114-and-D-type-retrovirus (RDR) interference group [99,100,101,102]. Recently, Suppressyn was found to function as an antiviral agent against RDR infections [8].

### 4.3. Other Placenta-Associated ERVs

In the human placenta, in addition to Syncytin-1 and -2, other ERV *env*-derived proteins are expressed, including ERVV-1, ERVV-2, ERV3-1 (also known as ERVR), and ERVMER34-1 (also known as HEMO). ERVV-1, ERVV-2, and ERVV3-1 are expressed in syncytiotrophoblasts throughout gestation [12,103,104]. Human ERVVV-1 and -2 should not have fusion activity [105,106]. However, ERVV-2 in old world monkeys (OWMs) and marmosets (a species of new world monkeys [NWMs]) shows fusion activity [106]. Thus, ERVV-2 is also called mac-syncytin-3 [106]. ERV3-1 has no fusion activity owing to a conserved premature stop codon but has a functional immunosuppressive domain [81,107,108]. ERVMER34-1 is expressed in cytotrophoblasts and extravillous trophoblasts in the first gestation and is detected in the blood of pregnant females [109]. ERVMER34-1 has no fusion activity, and its function is unknown; however, it localizes in a manner similar to ERVH48-1, suggesting that it may function as an inhibitor of cell fusion [12].

### 4.4. ERV Abnormalities and Placenta-Associated Diseases

In preeclampsia, which is one of the most severe complications during pregnancy, there is reported dysregulation of Syncytin-1 and -2 expression [14,110,111], leading to incomplete cell–cell fusion and abnormal differentiation from cytotrophoblasts to syncytiotrophoblasts [15,16]. IUGR, also known as fetal growth restriction, is a condition characterized by poor fetal development in the uterus during pregnancy. Syncytin-1 mRNA and protein expression levels are lower in the IUGR placenta than in normal placentas, resulting in abnormal cell fusion and increased apoptosis [17,18]. In addition, cell–cell fusion is disrupted in the placenta of trisomy 21 (Down syndrome) owing to the abnormal expression of Syncytin-2, resulting in dysplasia of the syncytiotrophoblast [112,113]. In trisomy 21 placenta, the number of Suppressyn-positive cytotrophoblasts increased, and the expression level of Suppressyn per cytotrophoblast also increased [19]. The expression level of Suppressyn in the serum is elevated in trisomy 21, resulting in excessive suppression of placental formation and immature placenta [19].

## 5. Acquisition and Exaptation of ERVs in Primates

The presence of retained ERV-FRD in apes, OWMs, and NWMs suggests that ERV-FRD infected and acquired these common ancestors over 40 MYA [106,114]. However, ERV-FRD Envelope proteins of NWMs have lost or have weakened cell–cell fusion ability [74,115] (Figure 3). Notably, NWMs have hemochorionic placentas and form syncytiotrophoblasts, which contradicts the dysfunction of ERV-FRD Envelope proteins in this species. This suggests that NWMs might possess another Syncytin-like factor. ERVV-1 and ERVV-2 are retained in apes, OWMs, and NWMs, suggesting that they were inserted into the genome of the common ancestor of Anthropoidea at the same time as ERV-FRD [116]. ERVV-2 may have originally had cell fusion activity, but it is speculated that this activity was lost in some apes and NWMs, suggesting that ERVV-2 may have been replaced by Syncytin-2 or may have acquired some other function, such as immunosuppression [106]. The ancestors of NWMs are believed to have migrated from Africa to South America approximately 34 MYA [117]; therefore, it is plausible that another Syncytin-like factor, such as ERVV-2, was acquired in these populations.

The retention of Syncytin-1 orthologs in apes and OWMs suggests that ERV-W infected the common ancestor of catarrhines after 40 MYA, following the divergence of catarrhines and platyrrhines. However, the open reading frame is disrupted by stop codons and frameshifts in OWMs [118], suggesting that exaptation, which refers to acquiring features or functions that are different from the original, occurred in the common ancestor of apes. Syncytiotrophoblasts are incapable of proliferating and retaining cellular activity when fused with cytotrophoblasts [63] (Appendix A). Moreover, Synytin-1 is expressed throughout gestation [73], suggesting that Syncytin-1 function is important for long-term placental maintenance. Around 24 MYA, apes and OWMs diverged, with larger apes prospering. Gestational length and body size are positively correlated [119]. Therefore, it is speculated that the acquisition of ERV-W enabled the maintenance of syncytiotrophoblastic trophoblasts, prolonged the gestation period, and increased ape size. The gestation period of great apes is 210 to 290 days, whereas that of OWMs is 150 days.

The presence of retained Suppressyn orthologs in apes and most OWMs suggests that ERVH48-1 was inserted into the common ancestor of catarrhines after 40 MYA [8,120]. This indicates that ERVH48-1 and ERV-W were simultaneously acquired. Both Syncytin-1 and Suppressyn are expressed during pregnancy but have different localization and expression dynamics. Syncytin-1 is expressed in syncytiotrophoblasts, and its expression level gradually increases during the differentiation process of primary cultured trophoblast cells [73,79,121]. Suppressyn is expressed in cytotrophoblast cells and EVT, and its expression level gradually decreases during the in vitro differentiation process [79]. The expression pattern of Suppressyn is more similar to that of ASCT2 than that of Syncytin-1 [79]. Although Suppressyn is known to have an antagonistic role against Syncytin-1, it was speculated that Suppressyn was acquired for other purposes. ERVH48-1 was also inactivated in some OWMs via frameshifting [8]. The ancestral sequence of Suppressyn retains its antiviral activity against RDR but is speculated to have lost its function in some lineages of OWMs. Suppressyn inhibits cell–cell fusion via Syncytin-1; however, it also inhibits infection via other retroviruses that use ASCT2 as a receptor [8]. The acquisition of the Suppressyn function is likely linked to the epidemic spread and conservation of some retroviruses that utilize ASCT2 as an antiviral factor. ASCT2, the receptor for Syncytin-1 and Suppressyn, contains eight transmembrane domains, and Region C at the C-terminus of extracellular loop 2 is identified as the host-range-determining region [95]. In mice and rats, insertions and deletions (indels) in Region C render ASCT2 non-functional as a receptor for RDR. These indels may have resulted in the acquisition of resistance to pathogenic viruses that utilize ASCT2 [122]. Comparing ASCT2 sequences in primates, a high degree of homology was observed among apes, OWMs, and NWM; however, some prosimians exhibited indels in Region C (Figure 4). ERVs utilizing ASCT2, including Syncytin-1 and Suppressyn, were not found in these species, suggesting that they may have acquired resistance to RDR via indels in Region C of ASCT2, as in mice and rats. Pig-tailed macaques also exhibit a large deletion in Region C; however, their genomes contain ERV-W and ERVH48-1 insertions, suggesting that this deletion occurred after the insertion of these viruses. It is speculated that Syncytin-1 and Suppressyn were acquired at around the same time, but a contradiction arises. Suppressyn inhibits cell–cell fusion by binding to ASCT2, the receptor for Syncytin-1. If retroviruses that use the same receptor cause receptor interference, it may be impossible for them to endogenize at the same time. The endogenization mechanism of retroviruses remains unclear. If the details of the acquisition process of Syncytin-1 and Suppressyn are revealed, it will be helpful to elucidate the endogenization mechanism.

ERV3-1 is conserved in apes except gorillas and in OWMs; it is thought to have entered the genome of the common ancestor of catarrhines after 40 MYA and was inactivated in gorillas [123].

ERVMER34-1 was retained in Laurasian and Eurasian theropods, but not in Afrotheria or Xenarthrans, suggesting that it entered the genome of Boreoeutheria at 100–120 MYA [109,124]. The ORF encoding the full-length protein was conserved in all simians and cats [109], suggesting that it may have some function.

## 6. Conclusions and Future Perspectives

The acquisition of the placenta is a crucial adaptation in mammals, which enables them to carry their developing fetuses within their bodies. The placenta protects the fetus from external threats and other risks. However, it also exposes the mother to other antigens and the risk of hemorrhage. Several retrotransposons and ERVs contributed to placental acquisition. For example, Syncytins were identified in various animal species as representative ERVs that contribute to placental formation. Syncytins have different origins in each animal species, likely arising from independent infection by other viruses; however, they acquired similar functions [11]. In primates, Suppressyn antagonizes Syncytin-1-mediated cell–cell fusion [94]. Syncytin-2 is thought to have infected and was inserted into the common ancestor of Anthropoidea, and Syncytin-1 and Suppressyn into the common ancestor of catarrhines. However, both were inactivated in some lineages, suggesting that multiple exaptation events had occurred. Syncytins and Suppressyn are species- or lineage-specific, posing challenges for in vivo analysis in mice, thereby impeding comprehensive analyses. Nevertheless, recent progress in research has enabled the isolation of trophoblast stem cells from human blastocysts and placentas in early pregnancy [125], and techniques were developed to induce the differentiation of trophoblast cells [126,127,128,129,130] and blastocyst-like tissue [131] from human pluripotent stem cells. These techniques will serve as valuable tools to further understand the function of ERVs in the placenta. The study of retrotransposons and ERVs, particularly Syncytins and Suppressyn, in placental development and function provides valuable insights into the genetic mechanisms underlying placental formation. Furthermore, understanding ERVs functioning in the placenta has potential implications for the management of placental-associated diseases such as preeclampsia, IUGR, and Down syndrome in humans.

## Figures and Tables

**Figure 1 biomolecules-13-01482-f001:**
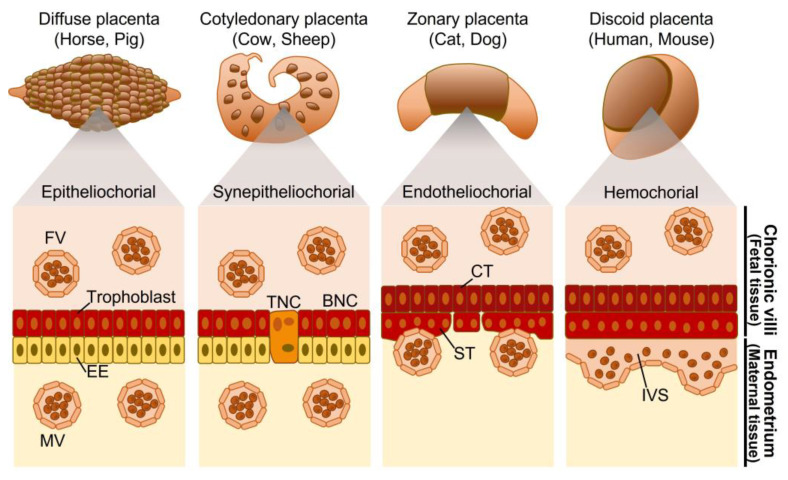
Placental classification based on the distribution of chorionic villi and tissue layer. FV, fetal blood vessel; MV, maternal blood vessel; EE, endometrial epithelium; BNC, binucleate cell; TNC, trinucleate cell; CT, cytotrophoblast; ST, syncytiotrophoblast; IVS, intervillous space.

**Figure 2 biomolecules-13-01482-f002:**
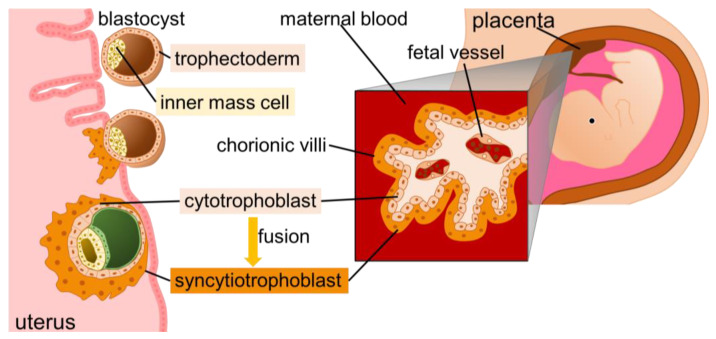
The process of placenta formation. Blastocysts are implanted in the endometrium, and trophectoderm differentiate into cytotrophoblasts. The cytotrophoblasts then fuse to form syncytiotrophoblasts, which are in direct contact with maternal blood.

**Figure 3 biomolecules-13-01482-f003:**
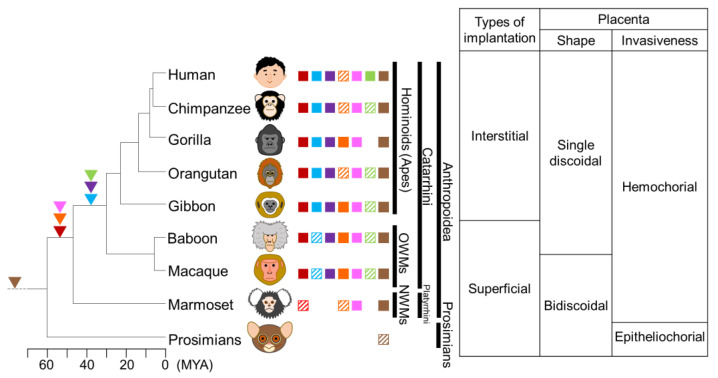
Phylogenetic tree of primates and status of placenta-associated ERVs with the type of implantation, shape, and invasiveness of placenta. Arrowheads represent the date of insertion of the indicated retroviral *env* genes into the genome of the primate ancestors. The filled squares indicate the conserved open reading frame of the *env* genes of ERVs, whereas the hatched squares indicate that the coding sequence was prematurely interrupted by mutations, insertions, deletions, and frameshifting. Red, blue, violet, orange, pink, green, and brown represent ERV-FRD, ERV-W, ERVH48-1, ERVV-1, ERVV-2, ERV3-1, and ERVMER34-1, respectively. MYA, million years ago; OWMs, Old World Monkeys; NWMs, New World Monkeys.

**Figure 4 biomolecules-13-01482-f004:**
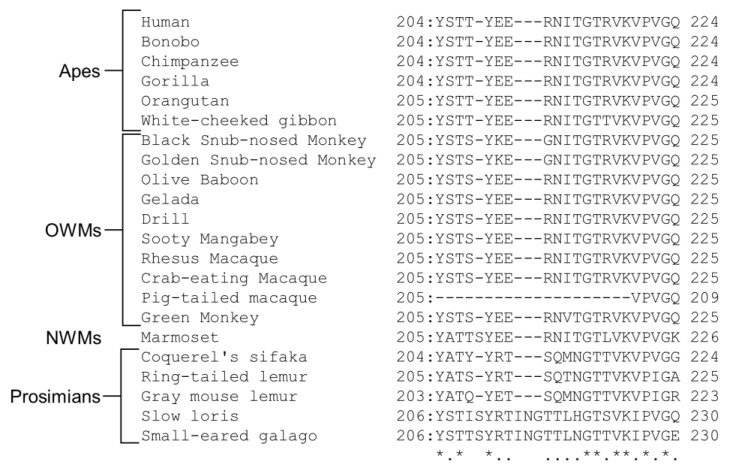
Multiple alignments of the amino acid sequences of region C of ASCT2 in Primates. The first and last numbers in the alignment indicate the start and end sites of each sequence, respectively. Amino acid sequences of alanine, serine, and cysteine transporter 2 (ASCT2) were obtained from the GenBank database with the following accession numbers: human (NM_005628.3), Bonobo (XM_003817486.4), Chimpanzee (XM_016936300.2), Gorilla (XM_019015032.2), Orangutan (XM_002829451.5), White-cheeked gibbon (XM_030807697.1), Black Snub-nosed Monkey (XM_017874300.1), Golden Snub-nosed Monkey (XM_030913729.1), Olive Baboon (XM_031659126.1), Gelada (XM_025365811.1), Drill (XM_011969657.1), Sooty Mangabey (XM_012080639.1), Rhesus Macaque (XM_015124323.2), Crab-eating Macaque (XM_045379941.1), Pig-tailed macaque (XM_011736052.2), Green Monkey (XM_037991882.1), Marmoset (XM_035284840.1), Coquerel’s sifaka (XM_012653384.1), Ring-tailed lemur (XM_045531338.1), Gray mouse lemur (XM_012761502.2), Slow loris (XM_053604173.1), and Small-eared galago (XM_012811197.2). Asterisks indicate conserved nucleotides between all species. Periods indicate sites with more than 50% matches.

## Data Availability

Not applicable.

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
