# Peer review of "Acquisition and Exaptation of Endogenous Retroviruses in Mammalian Placenta"

_biomolecules, 2023, doi:10.3390/biom13101482_

Round 1
Reviewer 1 Report
The manuscript from Sayumi Shimode presents a very interesting review. I appreciated reading this well presented and comprehensive manuscript.
I would only make minor comments as listed below, for the improvement of an expectedly published version.
Detailed comments:
Abstract:
-“ Several ERVs, such as Syncytins" : this should be “several ERV-encoded proteins….” Syncytins are not “ERVs” per se. Same comment for “investigate the role of ERVs, specifically Syncytin and Suppressyn”.
-protection from “weather” sound a little weird, makes it understandable as an “umbrella”. May be omitted?
-Suppresyn comes from nowhere. Should be introduced, at least along with syncytins?
-“ Here, we investigate the role of ERVs, specifically Syncytin(s) and Suppressyn” is it a research study (investigate) or a review? A better term would be more appropriate, or to be merged with “This review will provide….” ?
Keywords: Syncytin and Suppresyn should also be added.
Introduction:
-“Once retroviruses become integrated into the germline”, should rather be “when(ever) retroviruses become integrated into the germline” , since this is far from being the general case, rather exceptional but statistically significant over centuries or millennia. Also “they may result in an inherited proviral…” since only seldom occurrences, not always, but accumulating in time.
- « ……ERVs and retrotransposons to placental acquisition » Acquisition or, more globally, to the evolution of placental morphogenesis and physiology ?
2.1
-“ Moreover, eutherians possess » : rather « as a particular feature, eutherians” or “Eutherians additionally possess”
-“including weather and other exogenous…’ Here, may be mentioned as “weather threats” ?
2.2
-“ 40% of the eukaryotic genome contains retrotransposons” Rather “retrotransposons represent approximately 40% of the eukaryotic genome”
3. Classification of the mammalian placenta
- “Although they occupy a smaller area” ‘They” means placenta from primates ?.. The latter …..?
- “Although mammalian placentas are diverse, their structure and invasiveness do not 154 match the evolutionary phylogenetic tree of mammals, suggesting that they may have 155 evolved independently in each species while retaining their basic structure and function”. It should be noted here that this is a known phenomenon mediated by horizontal and inter-species/individuals transmission of ERV or retrotransposons copies via virion-producing copies from ERV families or other “carrier” viruses (Zhang, H.H., C. Feschotte, M.J. Han, and Z. Zhang. 2014. Recurrent horizontal transfers of Chapaev transposons in diverse invertebrate and vertebrate animals. Genome Biol Evol 6:1375-1386; Piskurek, O., and D.J. Jackson. 2012. Transposable elements: from DNA parasites to architects of metazoan evolution. Genes (Basel) 3:409-422; Gilbert, C. et al. 2010. A role for host-parasite interactions in the horizontal transfer of transposons across phyla. Nature 464:1347-1350; Belshaw, R., A. Katzourakis, J. Paces, A. Burt, and M. Tristem. 2005. High copy number in human endogenous retrovirus families is associated with copying mechanisms in addition to reinfection. Mol Biol Evol 22:814-817; Isfort, R., D. Jones, R. Kost, R. Witter, and H.J. Kung. 1992. Retrovirus insertion into herpesvirus in vitro and in vivo. Proc Natl Acad Sci U S A 89:991-995; etc.)
4. ERVs and placenta
-“ ERVs are retrovirus-like sequences ……and are passed on to offspring according to the Mendelian law.” This is in contradiction with all what is explained above, if understood as “ERVs” always follow the Mendelian rules. The very important and already mentioned (though it should be better highlighted and detailed) horizontal transmission between different phylogenic branches, should be taken into consideration here too, for they can generally be from Mendelian inheritance, except when resulting from horizontal transmission as described during evolution (cite relevant papers).
-“env gene in retroviruses encodes an envelope protein that binds to receptors on the host 169 cell surface and induces membrane fusion” …not always, not all of them.
- « Jerkzite sheep retrovirus (JSRV) » is Jaagsiekte SRV…
- “The Syncytin family comprises ERVs that function in the placenta…...” Should rather be: “Syncytins have functional roles in the placenta and are encoded by various ERV families… In humans, Syncytin-1 is encoded by the ERWE1 locus on Chr7q21.2…..” (Chromosomal location of primate synytin-1 related genes is not very well defines).
-“Syncytin-2 has immunosuppressive activity, whereas Syncytin-1 does not [85]” Cf. cited PNAS paper
-“ Single cell RNA sequencing data from the placenta at multiple developmental ….”
5. Acquisition and exaptation of ERVs in primates
-In all the text: HERV-W is for Human ERV-W. so, when writing about other species, please name it “ERV-W”.
6. Conclusions and future perspectives
- “Weather threats”, or so, would be more appropriate for a specific understanding, here again.
- “they acquired similar functions by chance”. It may be not necessary to bet on evolutionary drivers, and omit “by chance”.
Good. Few typos or inappropriate words are mentioned in the detailed comments.
Author Response
Thank you for your comments. We have revised the manuscript extensively according to your comments and suggestions.
Detailed comments:
Abstract:
-“ Several ERVs, such as Syncytins" : this should be “several ERV-encoded proteins….” Syncytins are not “ERVs” per se. Same comment for “investigate the role of ERVs, specifically Syncytin and Suppressyn”.
Answer: We have corrected the sentence accordingly (page 1, lines 11–12 in the revised version).
-protection from “weather” sound a little weird, makes it understandable as an “umbrella”. May be omitted?
Answer: We have omitted “weather” accordingly (page 1, line 13 in the revised version).
-Suppresyn comes from nowhere. Should be introduced, at least along with syncytins?
Answer: We have added text introducing Supressyn accordingly (page 1, lines 15–16 in the revised version).
-“ Here, we investigate the role of ERVs, specifically Syncytin(s) and Suppressyn” is it a research study (investigate) or a review? A better term would be more appropriate, or to be merged with “This review will provide….” ?
Answer: We have corrected the sentence accordingly (page 1, lines 19–20 in the revised version).
Keywords: Syncytin and Suppresyn should also be added.
Answer: We have added Syncytina and Suppressyn in Keywords accordingly (page 1, line 25 in the revised version).
Introduction:
-“Once retroviruses become integrated into the germline”, should rather be “when(ever) retroviruses become integrated into the germline” , since this is far from being the general case, rather exceptional but statistically significant over centuries or millennia. Also “they may result in an inherited proviral…” since only seldom occurrences, not always, but accumulating in time.
Answer: We have corrected the sentence accordingly (page 1, line 30 in the revised version).
- « ……ERVs and retrotransposons to placental acquisition » Acquisition or, more globally, to the evolution of placental morphogenesis and physiology ?
Answer: We have corrected the sentence accordingly (page 2, lines 49–50 in the revised version).
2.1
-“ Moreover, eutherians possess » : rather « as a particular feature, eutherians” or “Eutherians additionally possess”
Answer: We have corrected the sentence accordingly (page 2, lines 61–62 in the revised version).
-“including weather and other exogenous…’ Here, may be mentioned as “weather threats” ?
Answer: We have corrected the sentence accordingly (page 2, line 73 in the revised version).
2.2
-“ 40% of the eukaryotic genome contains retrotransposons” Rather “retrotransposons represent approximately 40% of the eukaryotic genome”
Answer: We have corrected the sentence accordingly (page 2, lines 87–88 in the revised version).
- Classification of the mammalian placenta
- “Although they occupy a smaller area” ‘They” means placenta from primates ?.. The latter …..?
Answer: “They” refers to placenta in primates. We have corrected the sentence to clarify this further (page 3, line 131 in the revised version).
- “Although mammalian placentas are diverse, their structure and invasiveness do not 154 match the evolutionary phylogenetic tree of mammals, suggesting that they may have 155 evolved independently in each species while retaining their basic structure and function”. It should be noted here that this is a known phenomenon mediated by horizontal and inter-species/individuals transmission of ERV or retrotransposons copies via virion-producing copies from ERV families or other “carrier” viruses (Zhang, H.H., C. Feschotte, M.J. Han, and Z. Zhang. 2014. Recurrent horizontal transfers of Chapaev transposons in diverse invertebrate and vertebrate animals. Genome Biol Evol 6:1375-1386; Piskurek, O., and D.J. Jackson. 2012. Transposable elements: from DNA parasites to architects of metazoan evolution. Genes (Basel) 3:409-422; Gilbert, C. et al. 2010. A role for host-parasite interactions in the horizontal transfer of transposons across phyla. Nature 464:1347-1350; Belshaw, R., A. Katzourakis, J. Paces, A. Burt, and M. Tristem. 2005. High copy number in human endogenous retrovirus families is associated with copying mechanisms in addition to reinfection. Mol Biol Evol 22:814-817; Isfort, R., D. Jones, R. Kost, R. Witter, and H.J. Kung. 1992. Retrovirus insertion into herpesvirus in vitro and in vivo. Proc Natl Acad Sci U S A 89:991-995; etc.)
Answer: Physiological and morphological evolution, which are not consistent with the molecular phylogenetic tree (often referred to as convergent evolution), are also caused by the horizontal and inter-species/individuals transmission of ERVs or retrotransposons copies via virion-producing copies from ERV families or other “carrier” viruses (Zhang et al., Genome Biol Evol., 2014; Piskurek et al., Genes (Basel), 2012; Gilbert et al., Nature, 2010; Belshaw et al., Mol Biol Evol., 2005; Isfort et al., PNAS., 1992). To date, multiple ERVs involved in placental diversification have been identified. Accordingly, we have discussed the horizontal transmission of ERVs and retrotransposons in the text (page 4, line 153–157 in the revised version).
- ERVs and placenta
-“ ERVs are retrovirus-like sequences ……and are passed on to offspring according to the Mendelian law.” This is in contradiction with all what is explained above, if understood as “ERVs” always follow the Mendelian rules. The very important and already mentioned (though it should be better highlighted and detailed) horizontal transmission between different phylogenic branches, should be taken into consideration here too, for they can generally be from Mendelian inheritance, except when resulting from horizontal transmission as described during evolution (cite relevant papers).
Answer: ERVs are retrovirus-like sequences that have been previously integrated into the host germline genome. ERVs are passed on to offspring by vertical transmission through Mendelian inheritance (Weiss, Retrovirology, 2006) and are spread between species and individuals by horizontal transmission (Belshaw et al., Mol Biol Evol., 2005). Accordingly, we have discussed the horizontal transmission of ERVs and retrotransposons in the text (page 4, lines 159–162 in the revised version).
-“env gene in retroviruses encodes an envelope protein that binds to receptors on the host 169 cell surface and induces membrane fusion” …not always, not all of them.
Answer: The two main routes of retrovirus entry into the host cells are direct membrane fusion and endocytosis (Dimitrov et al., Nat Rev Microbiol., 2004). When the envelope protein of the retrovirus binds to a specific receptor on cell surface, the envelope protein undergoes conformational change and direct membrane fusion occurs on the plasma membrane. In endocytosis, membrane fusion occurs in early endosomes after transport into the cytoplasm. EIAV, JSRV, and other retroviruses enter the host cells by endocytosis (Brindley et al., J Virol., 2008; Bertrand et al., J. Virol., 2007). Accordingly, we have discussed the horizontal transmission of ERVs and retrotransposons in the text (page 6, lines 226–231 in the revised version).
- « Jerkzite sheep retrovirus (JSRV) » is Jaagsiekte SRV…
Answer: We have corrected the spelling accordingly (page 6, line 234 in the revised version).
- “The Syncytin family comprises ERVs that function in the placenta…...” Should rather be: “Syncytins have functional roles in the placenta and are encoded by various ERV families… In humans, Syncytin-1 is encoded by the ERWE1 locus on Chr7q21.2…..” (Chromosomal location of primate synytin-1 related genes is not very well defines).
Answer: We have corrected the sentence accordingly (page 5, lines 192–193 in the revised version).
-“Syncytin-2 has immunosuppressive activity, whereas Syncytin-1 does not [85]” Cf. cited PNAS paper
Answer: We have corrected the sentence accordingly (page 6, line 205 in the revised version).
-“ Single cell RNA sequencing data from the placenta at multiple developmental ….”
Answer: We have corrected the sentence accordingly (page 7, line 260 in the revised version).
- Acquisition and exaptation of ERVs in primates
-In all the text: HERV-W is for Human ERV-W. so, when writing about other species, please name it “ERV-W”.
Answer: We have corrected the words accordingly.
- Conclusions and future perspectives
- “Weather threats”, or so, would be more appropriate for a specific understanding, here again.
Answer: We have omitted “weather” accordingly (page 10, line 394 in the revised version).
- “they acquired similar functions by chance”. It may be not necessary to bet on evolutionary drivers, and omit “by chance”.
Answer: We have omitted “by chance” accordingly (page 10, line 399 in the revised version).

Reviewer 2 Report
Two recent review articles (2021 and 2022) already address the link between ERV and placenta acquisition in mammals : (i) Endogenous Retroviruses and Placental Evolution, Development, and Diversity. Imakawa K et al. Cells. 2022 : 11(15):2458. doi: 10.3390/cells11152458 (ii) Syncytins expressed in human placental trophoblast. R Michael Roberts et al., Placenta 2021 : 113:8-14. doi: 10.1016/j.placenta.2021.01.006. The author of the submitted article does not cite these reviews. In my opinion, the present submitted article is no more informative than these two articles in the field.
Only paragraph 4.4 provides a new synthesis on the suppressyn genes (an Env ERV protein that confers protection from infection by other retroviruses and inhibit the synciytin-1 specific cell fusion). However, sincytin-1 and suppressyn genes were acquired by mammals in the same period of time and are co-expressed during the placenta formation, the author neither discusses this point nor the antagonistic role of these proteins in placenta formation.
Thus, I do not recommend the publication of this article in Biomolecules. Furthermore, the article should be rewritten taking into account many changes described below.
Major changes
Lane 34 : after [1-3], please add, « as well as in immune tolerance and resistance to infection by exogenous viruses by a receptor interference mechanism ».
Lane 43. Please add here that “syncitin proteins are encoded by modified env genes of endogenous retroviruses (ERV)”.
Lane 83: 40% of genome contain retrotransposons. Please change by “retrotransposons, transposons and repetitive sequences”
Lane 83-85. Please, change the sentence. Transposons relocate their genome by a cut and past mechanism. By contrast, retrotransposons duplicate their genome. The retrotransposon RNA is reverse-transcribed and the resulting DNA is inserted in another part of the genome.
Paragraph 2.2. PEG10 and PEG 11 have retroviral Gag and Pol encoding sequences. Do they encode Env proteins with fusion activity? It is not clear how they work in the placenta formation or acquisition.
Paragraph 3. This paragraph dealing with the classification of mammalian placenta is too long. It should be reduced or a modification of the article titer is required.
Paragraph 4.1. I do not understand the link of this paragraph dealing with receptor interfernce with the placenta formation. Please delete this paragraph or reduce it considerably and place it at the beginning of §4.4 relative to supressyn.
§4.3. To add in this §4.3. Syncytins are required for both cell fusion and proper placentation have been demonstrated in the mouse model (by knocking out genes equivalent to syncitins-1 and -2) with the relevant publication references.
Minor changes
Lane 48 : title of paragraph 2 is missing
Lane 43: please change “are implicated” by “are supposed to be implicated” in this diversity.
Lane 232. I guess it is “syncitin-1” that has not immunosuppressive activity
Lane 281 « suppressyn is located on chr 21 ». (already said previously)
Figure 4. could be placed in a supp data section. No real interest here
Lane 329. Please, replace « infected » by « acquired »
§4. Please add that a range of other ERV-like genes are expressed in human placental trophoblast such as ERVV-1, ERVV-2, and ERV3–1 as well as at later stages.
Author Response
Thank you for your comments. We have revised the manuscript extensively according to your comments and suggestions
Two recent review articles (2021 and 2022) already address the link between ERV and placenta acquisition in mammals : (i) Endogenous Retroviruses and Placental Evolution, Development, and Diversity. Imakawa K et al. Cells. 2022 : 11(15):2458. doi: 10.3390/cells11152458 (ii) Syncytins expressed in human placental trophoblast. R Michael Roberts et al., Placenta 2021 : 113:8-14. doi: 10.1016/j.placenta.2021.01.006. The author of the submitted article does not cite these reviews. In my opinion, the present submitted article is no more informative than these two articles in the field.
Answer: Accordingly, we have included the citations from the suggested review papers and have added information about other placenta-associated ERVs and the relationship between Syncytin-1 and Suppressyn.
Only paragraph 4.4 provides a new synthesis on the suppressyn genes (an Env ERV protein that confers protection from infection by other retroviruses and inhibit the synciytin-1 specific cell fusion). However, sincytin-1 and suppressyn genes were acquired by mammals in the same period of time and are co-expressed during the placenta formation, the author neither discusses this point nor the antagonistic role of these proteins in placenta formation.
Answer: Accordingly, we have discussed localization, antagonistic roles, and acquisition processes for Synsyctin-1 and Suppressyn in the revised manuscript.
Major changes
Lane 34 : after [1-3], please add, « as well as in immune tolerance and resistance to infection by exogenous viruses by a receptor interference mechanism ».
Answer: We have added the sentence accordingly (page 1, lines 35–36 in the revised version).
Lane 43. Please add here that “syncitin proteins are encoded by modified env genes of endogenous retroviruses (ERV)”.
Answer: We have added the sentence accordingly (page 1, line 46 in the revised version).
Lane 83: 40% of genome contain retrotransposons. Please change by “retrotransposons, transposons and repetitive sequences”
Answer: We have corrected the sentence accordingly (page 2, lines 87–88 in the revised version).
Lane 83-85. Please, change the sentence. Transposons relocate their genome by a cut and past mechanism. By contrast, retrotransposons duplicate their genome. The retrotransposon RNA is reverse-transcribed and the resulting DNA is inserted in another part of the genome.
Answer: We have corrected the sentence accordingly (page 2, lines 88–91 in the revised version).
Paragraph 2.2. PEG10 and PEG 11 have retroviral Gag and Pol encoding sequences. Do they encode Env proteins with fusion activity? It is not clear how they work in the placenta formation or acquisition.
Answer: PEG10 and PEG11 do not encode Env proteins with fusion activity. The Gag protein of PEG10 forms virus-like particles that bind to cellular RNAs including Hbegf (Heparin-binding EGF-like growth factor), playing an important role in placental formation (Abed et al., PLoS ONE, 2019). PEG10 might bind to and stabilize these cellular RNAs and has been suggested to be involved in placentation (Abed et al., PLoS ONE, 2019). The biochemical function of the Pol protein of PEG10 remains unknown; however, its protease domain is highly conserved in all Eutherians, suggesting that the protease activity plays an important role (Shiura et al., Development, 2021). The biochemical functions of PEG11 and Sirh7 are also unknown and require future investigation. Accordingly, we have discussed biochemical function and mechanism of PEG10 and PEG11 in the text (page 3, lines 100–104, 106–108, and 119–120 in the revised manuscript).
Paragraph 3. This paragraph dealing with the classification of mammalian placenta is too long. It should be reduced or a modification of the article titer is required.
Answer: We have reduced the length of paragraph 3 accordingly (page 3, line 123–page 4, line 157 in the revised version).
Paragraph 4.1. I do not understand the link of this paragraph dealing with receptor interfernce with the placenta formation. Please delete this paragraph or reduce it considerably and place it at the beginning of §4.4 relative to supressyn.
Answer: We have reduced the length of the paragraph on receptor interference and moved it to the beginning of section 4.4 (page 6, lines 226–242 in the revised manuscript).
- 4.3. To add in this §4.3. Syncytins are required for both cell fusion and proper placentation have been demonstrated in the mouse model (by knocking out genes equivalent to syncitins-1 and -2) with the relevant publication references.
Answer: We have added text on the knockout of Syncytins in a mouse model (page 6, lines 212–220 in the revised manuscript)
Minor changes
Lane 48 : title of paragraph 2 is missing
Answer: Accordingly, we have added the title of paragraph 2 (page 2, line 52 in the revised version).
Lane 43: please change “are implicated” by “are supposed to be implicated” in this diversity.
Answer: We have corrected the sentence accordingly (page 1, line 45 in the revised version).
Lane 232. I guess it is “syncitin-1” that has not immunosuppressive activity
Answer: We have corrected the sentence accordingly (page 6, line 205 in the revised version).
Lane 281 « suppressyn is located on chr 21 ». (already said previously)
Answer: We have deleted the sentence accordingly (page 7, line 292 in the revised version).
Figure 4. could be placed in a supp data section. No real interest here
Answer: Figure 4 has been moved to the supplemental material accordingly (Figure S1).
Lane 329. Please, replace « infected » by « acquired »
Answer: We have replaced the word accordingly (page 8, line 335 in the revised version).
- 4. Please add that a range of other ERV-like genes are expressed in human placental trophoblast such as ERVV-1, ERVV-2, and ERV3–1 as well as at later stages.
Answer: In the human placenta, in addition to ERVW-1 and ERVFRD-1, other ERV env-derived proteins expressed, including ERVV-1, ERVV-2, ERV3-1 (also known as ERVR), and ERVMER34 (also known as HEMO). ERVV-1, ERVV-2, and ERVV3-1 are expressed in syncytiotrophoblasts throughout gestation (Boyd et al., Virology, 1993; West et al., PNAS., 2019; Roberts et al., Placenta, 2021). Human ERVVV-1 and ERVVV-2 are not supposed to have fusion activity (Blaise et al., Retrovirology, 2005; Esnault et al., PLoS Genet., 2013). However, ERVV-2 in old world monkeys (OWMs) and marmosets (a species of new world monkeys [NWMs]) showed fusion activity (Esnault et al., PLoS Genet., 2013). Thus, ERVV-2 is also called mac-syncytin-3 (Esnault et al., PLoS Genet., 2013). ERV3-1 has no fusion activity owing to a conserved premature stop codon but has a functional immunosuppressive domain (Venables et al., Virology, 1995; Matsuda et al., Oncogene 1997; Mangeney et al., PNAS., 2007). ERVMER34-1 is expressed in cytotrophoblasts and extravillous trophoblast in the first gestation and is detected in the blood of pregnant women (Heidmann et al. PNAS., 2017). ERVMER34-1 has no fusion activity and its function is unknown, but it localizes in a manner similar to ERV-Fb1, suggesting that it may function as an inhibitor of cell fusion (Roberts et al., Placenta, 2021).
ERVV-1 and ERVV-2 are retained in apes, OWMs, and NWMs, suggesting that they were inserted into the genome of the common ancestor of Anthropoidea at the same time as ERV-FRD (Kjeldbjerg et al., BMC Evol Biol., 2008). ERVV-2 may have originally had cell fusion activity, but it is speculated that this activity was lost in some apes and NWMs, suggesting that ERVV-2 may have been replaced by Syncytin-2 or may have acquired some other function such as immunosuppression (Esnault et al., PLoS Genet., 2013). ERV3-1 is conserved in apes except gorilla and in OWMs; it is thought to have entered the genome of the common ancestor of catarrhines after 40 MYA and was inactivated in gorillas (Herve´ et al. Genomics, 2003). ERVMER34-1 was retained in Laurasian and Eurasian theropods, but not in Afrotheria or Xenarthrans, suggesting that it entered the genome of Boreoeutheria at 100–120 MYA (Meredith et al., Science, 2011; Heidmann et al., PNAS., 2017). The ORF encoding the full-length protein was conserved in all apes and cats (Heidmann et al., PNAS., 2017), suggesting that it may have some function.
Accordingly, we have discussed expression patterns, function, and evolutional history of other placenta-associated ERVs in the text and Figure 3 (page 7, lines 270-282, page 8, lines 301–306, page 9, lines 369–371, page 10, lines 372–375 in the revised version).
Round 2
Reviewer 2 Report
The authors have made a real effort to respond to all my comments and the manuscript seems acceptable to me as it currently stands.However, they still have to properly add the two references requested previously:
The first one (ndogenous Retroviruses and Placental Evolution, Development, and Diversity. Imakawa K et al. Cells. 2022 : 11(15):2458) is not correctly cited (reference 46 in the placenta section is not appropriate). Please note that the page number (2458) is missing in the reference section.
I did not find the other requested citation (Syncytins expressed in human placental trophoblast. R Michael Roberts et al., Placenta 2021 : 113:8-14).
I strongly suggest including them in the introductory section.Author Response
Thank you for your comments. We have revised the manuscript extensively according to your comments and suggestions.
The authors have made a real effort to respond to all my comments and the manuscript seems acceptable to me as it currently stands.
However, they still have to properly add the two references requested previously:
The first one (ndogenous Retroviruses and Placental Evolution, Development, and Diversity. Imakawa K et al. Cells. 2022 : 11(15):2458) is not correctly cited (reference 46 in the placenta section is not appropriate). Please note that the page number (2458) is missing in the reference section.
I did not find the other requested citation (Syncytins expressed in human placental trophoblast. R Michael Roberts et al., Placenta 2021 : 113:8-14).
I strongly suggest including them in the introductory section.
Answer: Accordingly, we have included the citations from the suggested review papers in introduction section and corrected the reference list.